Spatiotemporal expansion of human brucellosis in Shaanxi Province, Northwestern China and model for risk prediction

Yang Zurong 1 2
Pang Miaomiao 3
Zhou Qingyang 2
Song Shuxuan 1
Liang Weifeng 4
Chen Junjiang 1
Guo Tianci 1
Shao Zhongjun 13759981783@163.com 1
Liu Kun liukun5959@qq.com liukun5959@fmmu.edu.cn 1
1 Department of Epidemiology, Ministry of Education Key Lab of Hazard Assessment and Control in Special Operational Environment, School of Public Health, Air Force Medical University , Xi’an , Shaanxi , People’s Republic of China
2 Centre for Disease Prevention and Control in Northern Theater Command , Shenyang , People’s Republic of China
3 Shaanxi Provincial Corps Hospital of Chinese People’s Armed Police Force , Xi’an , Shaanxi , People’s Republic of China
4 Health Commission of Shaanxi Province , Xi’an , Shaanxi , People’s Republic of China
Driscoll Timothy
Electronic publication date: 2020 Oct 19
Publication date: 2020
Volume: 8
Electronic Location ID: e10113
Received 2020 Jun 5; Accepted 2020 Sep 16
Copyright: ©2020 Yang et al.
Copyright year: 2020
Copyright holder: Yang et al.
License: This is an open access article distributed under the terms of the Creative Commons Attribution License, which permits unrestricted use, distribution, reproduction and adaptation in any medium and for any purpose provided that it is properly attributed. For attribution, the original author(s), title, publication source (PeerJ) and either DOI or URL of the article must be cited.
License URL: https://creativecommons.org/licenses/by/4.0/

Keywords: Human brucellosis, Spatiotemporal expansion, Generalized additive mixed model, Regional differences, Meteorological factors

Funding: National Natural Science Foundation of China 81803289 Natural Science Foundation of Shaanxi Province 2020JM-329 This work was supported by the National Natural Science Foundation of China (No. 81803289), the Natural Science Foundation of Shaanxi Province (No. 2020JM-329). The funders had no role in study design, data collection and analysis, decision to publish, or preparation of the manuscript.

==============================
Background

Human brucellosis imposes a heavy burden on the health and economy of endemic regions. Since 2011, China has reported at least 35,000 human brucellosis cases annually, with more than 90% of these cases reported in the northern. Given the alarmingly high incidence and variation in the geographical distribution of human brucellosis cases, there is an urgent need to decipher the causes of such variation in geographical distribution.

Method

We conducted a retrospective epidemiological study in Shaanxi Province from January 1, 2005 to December 31, 2018 to investigate the association between meteorological factors and transmission of human brucellosis according to differences in geographical distribution and seasonal fluctuation in northwestern China for the first time.

Results

Human brucellosis cases were mainly distributed in the Shaanbei upland plateau before 2008 and then slowly extended towards the southern region with significant seasonal fluctuation. The results of quasi-Poisson generalized additive mixed model (GAMM) indicated that air temperature, sunshine duration, rainfall, relative humidity, and evaporation with maximum lag time within 7 months played crucial roles in the transmission of human brucellosis with seasonal fluctuation. Compared with the Shaanbei upland plateau, Guanzhong basin had more obvious fluctuations in the occurrence of human brucellosis due to changes in meteorological factors. Additionally, the established GAMM model showed high accuracy in predicting the occurrence of human brucellosis based on the meteorological factors.

Conclusion

These findings may be used to predict the seasonal fluctuations of human brucellosis and to develop reliable and cost-effective prevention strategies in Shaanxi Province and other areas with similar environmental conditions.

Introduction

Brucellosis is a bacterial zoonosis caused by genus Brucella, including B. abortus, B. canis, B. suis, B. ceti, B. pinnipedialis and B. inopinata (Dadar, Shahali & Whatmore, 2019). Compared with the individual-to-individual transmission, the environment-to-individual transmission is more common (Li et al., 2017), which means that most people are infected by contacting with Brucella in the environment, such as contaminated forage, water, grass, liquids, products, raw milk and the uterine fluids from infected animals (Li et al., 2017; Nematollahi et al., 2017; Chen et al., 2013; Liu et al., 2020). Thus, shepherds, breeders, abattoirs workers, veterinarians and laboratory personnel who potentially contact the bacteria are at high risks (Chen et al., 2016). Infected people present a variety of symptoms such as night sweats, arthralgia, undulant fever, hepatomegaly, headaches, myalgia, and personality changes (Wang et al., 2020; Lou et al., 2016; Li et al., 2013). Since most infected people go to the clinic for treatment only when they have clinical symptoms such as undulant fever, asymptomatic infections often missed and/or misdiagnosed (Zhen et al., 2013). In addition, low awareness, ineffective preventative measures, high initial treatment failure, substantial residual disability, and relapse rates have contributed to the heavy burden of this disease on the health and economy of endemic regions (Lai et al., 2017).

Globally, the World Health Organization (WHO) and the Food and Agriculture Organization of the United Nations (FAO) announced that brucellosis is one of the most significantly neglected zoonotic diseases in the world (Dadar, Shahali & Whatmore, 2019). The virus results in tremendous economic losses and health threats in countries whose economies are dominated by livestock keeping. Human brucellosis is epidemic in many countries especially in Latin America, Middle East, and South and Central Asia, with a total of cases more than 500,000 cases each year (Zhao et al., 2019). Since 2011, 31 provinces in China have reported at least 35,000 human brucellosis cases annually, with more than 90% of these cases reported in northern China (Wang et al., 2013). Nationwide, the incidence of human brucellosis shows an apparent geographic expansion from northern pastureland provinces to the adjacent grassland and agricultural areas, then to southern coastal and southwestern areas (Guan, Wu & Huang, 2018; Yang et al., 2020). From 2005 to 2018, a total of 12,671 confirmed human brucellosis cases were reported in Shaanxi Province, with approximately 70.36% of cases reported in Shaanbei upland plateau, and the average annual incidence reached 11.50/100,000, which was higher than the national average incidence in 2018 (2.73/100,000) (Peng et al., 2020). Given the alarmingly high incidence and variation in the geographical distribution of human brucellosis cases, there is an urgent need to decipher the causes of such variation in geographical distribution.

The aim of the current study was to examine the spatial and temporal distributions of human brucellosis in Shaanxi Province between 2005 and 2018. We also explored the associations between meteorological factors and the environment-to-individual transmission of human brucellosis using a quasi-Poisson generalized additive mixed model (GAMM), and then analyzed the driving effect of meteorological factors on the distribution of human brucellosis and predicted the short-term incidence trend in the main epidemic areas of Shaanxi, northwestern China.

Materials & Methods

Study region

Shaanxi is a northwestern administrative province of China, extending from 31°42′–39°35′N to 105°29′–115°15′E. It consists of 11 prefecture-level cities including 30 municipal districts and 77 counties with an area of 205,800 km2 and a total population of 3.86 million in 2018 (http://tjj.shaanxi.gov.cn/index.htm). According to the typical climate and landforms, Shaanxi Province can be divided into three distinct natural sub-regions: the Shaanbei upland plateau (northern Shaanxi), the Guanzhong basin (middle part of Shaanxi), and the Shaannan mountainous region (southern Shaanxi) (Fig. S1). The Shaanbei upland plateau includes two cities (Yulin and Yan’an) and is part of the transitional landscape spanning from the Maowusu Desert to the Loess Plateau, which belongs to a typical crisscross zone of livestock keeping and farming. This region has constantly been one of the most severely affected areas by brucellosis in China (Chen et al., 2013), and the nearby provinces (Inner Mongolia, Gansu, Ningxia, and Shanxi) have all reported a high prevalence of brucellosis (Lai et al., 2017; Wang et al., 2013). The Guanzhong basin is located in the middle of Shaanxi province and mainly comprises plains and river landforms. This region has a large population with a huge demand for milk and meat products; therefore, transportation and livestock trade are active in this region. In recent years, the epidemic of brucellosis showed a rapid increasing trend and mainly affected people with occupational exposures, such as farmers, market workers, slaughterhouse workers, and veterinarians (Chen et al., 2016; Zhao et al., 2019). The Shaannan mountainous region is characterized by mountains and hills with a subtropical and continental monsoon climate. Sporadic human brucellosis cases have been reported in the region.

Data collection

In China, human brucellosis is listed as a class B notifiable infectious disease, and reporting of data for diagnosed cases (including age, gender, occupation, address and date of brucellosis onset) to the local Center for Disease Control and Prevention (CDC) through the National Notifiable Infectious Diseases Reporting Information System is mandatory. For this study, we obtained the data of human brucellosis cases in Shaanxi from January 1, 2005 to December 31, 2018 from the CDC of Shaanxi Province. All cases were diagnosed based on a combination of epidemiologic exposures, clinical features (fever lasting several days or weeks, sweating, fatigue, and muscle or joint pain, e.g.), and serological test results. Confirmatory tests including the standard plate agglutination test (PAT), rose bengal plate test, serum agglutination test, or isolation of Brucella spp were further conducted to confirm the human brucellosis cases. In addition, the demographic data from the 6th census conducted by the National Bureau of Statistics of China in 2010 were used to calculate the incidence of human brucellosis. The local monthly meteorological data for air temperature (°C), evaporation (mm), rainfall (mm), sunshine duration (h) and relative humidity (%) for the study period were obtained from the Chinese Bureau of Meteorology (http://data.cma.cn/).

Statistical analysis

We determined the spatiotemporal distributions of human brucellosis in Shaanxi Province between 2005 and 2018 with stratification for three sub-regions, and the monthly accumulative incidence of brucellosis was geo-referenced to each county on a digital map. In addition, we used a time-series analysis method to identify the relationships and potential effects between meteorological factors and the incidence of human brucellosis. Specifically, we utilized the cross-correlation analysis to assess the associations between meteorological factors and human brucellosis for a range of lags up to 7 months and presented meteorological factors with the maximum correlation coefficients in areas with high incidences (Shaanbei upland plateau and Guanzhong basin). Finally, to identify potential non-linear relationships between meteorological factors and the monthly incidence of human brucellosis, we applied cubic spline analysis including those variables from the quasi-Poisson GAMM model; the model was performed to examine the independent contribution of meteorological factors to the transmission of human brucellosis (Cao et al., 2020; Sun et al., 2018). To adjust for long-term trends, seasonality, and over-dispersion, we selected an appropriate degree of freedom (df) and lag for each variable in the model (Duan et al., 2017; Zhang et al., 2010). The structure of the model used in the current study is shown below: (1) logYt=β0+Group+sincidence,month+∑svariablest−e,by=Group,df,

where Yt is the number of human brucellosis cases in month t; β0 is the intercept; Group is used to control regional difference; sincidence,month exploits the link between incidence and month to control random effects; ∑svariablest−e,by=Group,df denotes the cubic spline of meteorological factors, including air temperature, evaporation, rainfall, sunshine duration and relative humidity, in the previous e months with corresponding df = 12 for different region. The R squared value, deviance explained value, and generalized cross-validation (GCV) estimation value were used to determine the most appropriate model.

The statistical analyses were performed using R software version 3.6.0 with the packages of “mgcv” and “itsadug”. The spatial analyses were performed using ArcGIS 10.2 Software (ESRI Inc.; Redlands, CA, USA). All statistical tests were two-sided, and a p value <0.05 was considered statistically significant.

Results

Spatial and temporal distributions of human brucellosis

Between 2005 and 2018, a total of 12,671 confirmed human brucellosis cases were reported in Shaanxi Province with significant geographical heterogeneity. Nearly 70.36% of these cases (8,916) were reported in Shaanbei upland plateau, and approximately 28.06% cases (3,555) were reported in Guanzhong basin. Only 200 cases (1.58%) were reported in Shaannan mountainous region. The annual number of cases of human brucellosis in Shaanxi fluctuated from 585 (1.57 per 100,000 persons) in 2005 to 673 (1.80 per 100,000 persons) in 2018, with two peaks in 2008 (1,238 cases, 3.32 per 100,000 persons) and 2014 (1,542 cases, 4.13 per 100,000 persons) (Fig. 1). When stratified by the three sub-regions, the annual number of cases in Shaanbei upland plateau showed bimodal peaks with a major peak in 2008 (1,104 cases, 19.93 per 100,000 persons) and a minor peak in 2014 (794 cases, 14.34 per 100,000 persons); however, the annual number of cases of Guanzhong basin gradually increased from 2008, peaking in 2014 with 729 cases (3.12 per 100,000 persons). The annual number of cases of Shaannan mountainous region slowly increased as well, peaking in 2015 with 83 cases (0.99 per 100,000 persons) (Fig. 2). The spatial distribution of human brucellosis showed that most cases occurred in Shaanbei upland plateau before 2008, with gradual expansion to middle and southern region (Fig. 3A). Notably, Dali county and Chengcheng county, which were located in low incidence areas, were sites of outbreaks in 2014. The monthly accumulative incidence showed seasonal fluctuations (Fig. 3B), with 60.36% (7,648) of cases occurring between March and July. Sporadic human brucellosis cases occurred in numerous regions of Shaannan mountainous region without significant seasonal fluctuations. Further analysis found that the peak of human brucellosis in Shaanbei upland plateau occurred in March to July, but Guanzhong basin occurred in April to July (Figs. 3C and 3D). In addition, Zizhou County reported the highest annual incidence (153.46 per 100,000 persons) in 2008 and the highest monthly accumulative incidence (104.03 per 100,000 persons) in April.

Figure 1 Temporal distribution of human brucellosis in Shaanxi Province, China, 2005–2018.

The bar chart represents the monthly incidence of human brucellosis in Shaanxi, and the line represents the annual number of cases of human brucellosis in Shaanxi Province.

Figure 2 Temporal distribution of human brucellosis in Shaanbei upland plateau, Guanzhong basin, and Shaannan mountainous region, 2005–2018.

Figure 3 Spatial and temporal distribution of human brucellosis in Shaanxi Province.

(A) Annual incidence of human brucellosis in each county in Shaanxi Province, 2005–2018. The map was created by Zurong Yang in ArcGIS 10.1 Software, ESRI Inc., Redlands, CA, USA, (https://www.arcgis.com/index.html). (B) Spatial distribution of human brucellosis in Shaanxi Province from January to December, 2005–2018. The map was created by Zurong Yang in ArcGIS 10.1 Software, ESRI Inc., Redlands, CA, USA, (https://www.arcgis.com/index.html). (C) Seasonal distribution of human brucellosis in Shaanbei upland plateau. (D) Seasonal distribution of human brucellosis in Guanzhong basin.

Correlations between human brucellosis incidence and meteorological factors

Shaanbei upland plateau and the Guanzhong basin had the highest numbers of human brucellosis cases in Shaanxi as major endemic area. Figures S2A & S2B showed the monthly incidence of human brucellosis in Shaanbei upland plateau was periodic and seasonal fluctuations. In contrast, the periodic and seasonal fluctuations of monthly incidence in Guanzhong basin gradually became obvious, as the incidence increased. The mean air temperature, sunshine duration, evaporation illustrated showed a similar fluctuation with the monthly incidence, while for rainfall and relative humidity an opposite trend was observed. Therefore, correlations between meteorological factors and the number of human brucellosis cases were further explored in these two regions (Table 1). The maximum correlation coefficients for meteorological factors, including air temperature, rainfall, relative humidity, sunshine duration and evaporation, showed lag time of 4 months, 3 months, 2 months, 6 months, and 5 months in Shaanbei upland plateau, respectively. However, the maximum correlation coefficients for the abovementioned meteorological factors showed lag time of 5 months, 4 months, 3 months, 7 months, and 6 months in the Guanzhong basin, respectively, which all lagged 1 month behind those in Shaanbei upland plateau. According to our correlation analysis between meteorological factors and monthly number of reported cases in the two regions, air temperature had a relatively stronger correlation than other factors, with Spearman correlation coefficients of 0.56 and 0.36 in Shaanbei upland plateau and the Guanzhong basin, respectively (Table 1 and Figs. S3A & S3B). The mean values for the monthly number of human brucellosis cases and meteorological factors in Shaanbei upland plateau and the Guanzhong basin are presented in Table 2. From 2005 to 2018 in Shaanbei upland plateau, the mean values for monthly cases, air temperature, relative humidity, cumulative rainfall, cumulative evaporation, and cumulative sunshine duration were 53.07 cases, 9.90 °C, 55.53%, 40.78 mm, 122.86 mm, and 215.87 h, respectively. The corresponding mean values in the Guanzhong basin were 21.16 cases, 12.10 °C, 65.16%, 50.79 mm, 101.97 mm, and 171.90 h, respectively.

Table 1 Correlation analysis between meteorological factors and the number of human brucellosis cases in Shaanbei upland plateau and the Guanzhong basin in Shaanxi Province, China, 2005–2018.

Meteorological factors	Shaanbei upland plateau	Guanzhong basin	
	Spearman coefficient	Lag value (month)	p	Spearman coefficient	Lag value (month)	p	
Air temperature	0.56	0	<0.001 **	0.36	0	<0.001**	
Rainfall	0.38	0	<0.001 **	0.25	0	0.010*	
Evaporation	0.67	0	<0.001 **	0.30	0	<0.001**	
Sunshine duration	0.54	0	<0.001 **	0.20	0	0.010*	
Humidity	−0.22	0	0.01 *	−0.06	0	0.460	
Lag in air temperature	−0.75	4	<0.001 **	−0.40	5	<0.001**	
Lag in rainfall	−0.59	3	<0.001	−0.34	4	<0.001**	
Lag in evaporation	−0.68	5	<0.001**	−0.37	6	<0.001**	
Lag in sunshine duration	−0.50	6	<0.001 **	−0.31	7	<0.001**	
Lag in humidity	−0.66	2	<0.001 **	−0.35	3	<0.001**	
Notes.

* p < 0.05.

** p < 0.001.

Table 2 Summary of monthly numbers of human brucellosis cases and meteorological factors in Shaanbei upland plateau and the Guanzhong basin in Shaanxi Province, China, 2005–2018.

	Variables	Min	P5	P25	P50	P75	P95	Max	Mean ± SD	
Shaanbei upland plateau	Number of cases	9.00	18.00	29.25	44.50	68.00	121.00	164.00	53.07 ± 31.70	
Mean air temperature (°C)	−10.86	−6.51	0.74	11.39	19.64	23.64	25.00	9.90 ± 10.40	
Mean relative humidity (%)	28.88	34.54	46.71	54.74	65.96	74.24	79.36	55.53 ± 12.34	
Rainfall (mm)	0.00	0.35	6.89	22.30	60.35	124.67	301.60	40.78 ± 45.41	
Evaporation (mm)	25.39	35.16	65.32	119.17	174.18	224.96	258.29	122.86 ± 62.57	
Sunshine duration (h)	106.44	139.14	190.46	219.22	244.82	279.98	292.00	215.87 ± 40.87	
Guanzhong basin	Number of cases	0.00	1.00	7.00	15.00	30.75	57.55	111.00	21.16 ± 19.92	
Mean air temperature (°C)	−5.51	−2.07	3.92	13.23	21.08	24.64	26.55	12.10 ± 19.92	
Mean relative humidity (%)	39.37	46.77	57.43	64.48	73.08	82.15	86.34	65.16 ± 10.51	
Rainfall (mm)	0.03	1.19	10.27	36.98	80.56	137.75	297.17	50.79 ± 48.77	
Evaporation (mm)	27.33	36.72	57.32	94.73	140.94	189.66	269.21	101.97 ± 49.46	
Sunshine duration (h)	48.11	90.09	138.38	170.67	204.59	249.34	285.13	171.90 ± 45.81	
Notes.

Min minimum of the variable

P5 the 5th percentile of the variable

P25 the 25th percentile of the variable

P50 the 50th percentile of the variable

P75 the 75th percentile of the variable

P95 the 95th percentile of the variable

Max maximum of the variable

SD standard deviation

Estimation of meteorological effects on human brucellosis in Shaanbei upland plateau and Guanzhong basin

To analyze the seasonal fluctuation of human brucellosis of Shaanbei upland plateau and the Guanzhong basin, a quasi-Poisson GAMM model was used. After controlling for auto-correlation, seasonality, and the lag effect, we found that the monthly number of human brucellosis cases was significantly associated with previous cases and meteorological factors (most p values <0.05), including air temperature, relative humidity, rainfall, evaporation, and sunshine duration (Table 3). According to the values for R square (0.90), deviance explained (91.00%), and GCV principles (4.32), the most appropriate GAMM model was fitted and selected. The observed and fitted cases from the final model matched relatively well for Shaanbei upland plateau and the Guanzhong basin, and the respective goodness of fit (R square) values were 91.43% and 87.83%, respectively. A good fit between observed cases and predicted cases was achieved, using the 24-month observations, and the goodness-of-fit analyses showed that the residuals did not cause significant auto-correlation in the final model (Figs. 4A & 4B and 5A & 5B). In addition, we found the effects of meteorological factors on the occurrence of human brucellosis were different in Shaanbei upland plateau and Guanzhong basin through Figs. 6A–6J and Table 3. Compared with the Shaanbei upland plateau, Guanzhong basin had more obvious fluctuations in the occurrence of human brucellosis due to changes in meteorological factors, such as air temperature, relative humidity, rainfall, sunshine duration, and evaporation. The summed effects of the meteorological factors parameters terms in Guanzhong basin was 3.48 (95% CI [2.84–4.12]), while the Shaanbei upland plateau was 2.94 (95% CI [2.70–3.20]). The effect of the meteorological factors indicated that except for the air temperature of the current month and rainfall of the previous month showed an upward trend, and previous sunshine duration was U-shaped, the rest of the meteorological factors was a downward trend in Guanzhong basin. High air temperature, low relative humidity, less rainfall, short sunshine duration, and suitable evaporation in this region all had positive effects on the incidence of human brucellosis. In addition, such positive effects were observed for low air temperature and long sunshine duration of the previous month. However, only a weak trend could be observed in Shaanbei upland plateau. Furthermore, the identified interactions between two meteorological factors in association with human brucellosis were shown in Fig. S4A–S4J. The result showed the interaction between high air temperature and lower relative humidity, higher air temperature and shorter sunshine duration, lower relative humidity and shorter sunshine duration, lower relative humidity and suitable evaporation, and less rainfall and suitable evaporation were all obviously associated with high incidence of human brucellosis. Especially in the environment of high air temperature, low relative humidity and short sunshine duration, the risk of human brucellosis was higher.

Table 3 Approximate significance of smooth terms in Shaanbei upland plateau and Guanzhong basin in Shaanxi Province, China, 2005–2018.

Variables	Regions	Effective degrees of freedom	F	p	
s(Air temperature)	Guanzhong basin	6.41	1.80	0.08	
Shaanbei upland plateau	1.00	0.86	0.35	
s(Lag in air temperature)	Guanzhong basin	7.73	1.49	0.15	
Shaanbei upland plateau	1.59	4.34	0.01	
s(Relative humidity)	Guanzhong basin	5.44	3.61	<0.001	
Shaanbei upland plateau	1.00	0.12	0.73	
s(Lag in relative humidity)	Guanzhong basin	6.62	4.06	<0.001	
Shaanbei upland plateau	1.00	4.21	0.04	
s(Sunshine duration)	Guanzhong basin	2.37	2.71	0.04	
Shaanbei upland plateau	1.75	3.18	0.04	
s(Lag in sunshine duration)	Guanzhong basin	9.41	1.66	0.09	
Shaanbei upland plateau	1.00	4.64	0.03	
s(Evaporation)	Guanzhong basin	3.47	4.60	<0.001	
Shaanbei upland plateau	2.31	1.63	0.21	
s(Lag in evaporation)	Guanzhong basin	1.30	6.37	0.01	
Shaanbei upland plateau	1.00	4.01	0.05	
s(Rainfall)	Guanzhong basin	8.61	2.38	0.01	
Shaanbei upland plateau	1.00	0.12	0.73	
s(Lag in rainfall)	Guanzhong basin	8.23	1.75	0.08	
Shaanbei upland plateau	1.00	0.42	0.52	
Notes.

* Summary of the GAMM smooth terms, their p-values and effective degrees of freedom are listed. We constructed the final model based on current monthly meteorological factors, temperature lag of 4 months, rainfall lag of 3 months, relative humidity lag of 2 months, sunshine duration lag of 6 months and evaporation lag of 5 months in Shaanbei upland plateau. Meanwhile, we constructed the final model for the Guanzhong basin based on current monthly meteorological factors and previous meteorological factors with lag times 1 month greater than those in Shaanbei upland plateau.

Discussion

To our best knowledge, this is the first comprehensive study to analyze the association between meteorological factors and transmission of human brucellosis according to differences in geographical distribution and seasonal fluctuation. We found that human brucellosis cases were distributed mainly in the northern part of Shaanxi province before 2008 and then slowly extended towards south with obvious seasonal fluctuations. The quasi-Poisson GAMM model suggested that air temperature, sunshine duration, rainfall, relative humidity, and evaporation with lag time within 7 months may play a crucial role in the transmission, especially environment-to-individual transmission, of human brucellosis and the variation in geographical distribution, and the model had great accuracy in predicting the occurrence of human brucellosis. The results suggested that the established GAMM can accurately forecast the short-term incidence over 24 months based on meteorological factors, which has important public health implications for developing reliable and cost-effective prevention strategies, including vaccination time, reservoir surveillance, environment disinfection frequency, elimination rates of infected animals and medical resource allocation.

The higher incidences of human brucellosis in Shaanbei upland plateau corroborated with the observation that the northern part of China has historically experienced severe endemic area of brucellosis (Wang et al., 2013). In fact, the cases of human brucellosis reported in the northern part of China account for more than 90% of the total reported cases every year (Chen et al., 2013; Lai et al., 2017; Wang et al., 2013; Shi et al., 2017). Our study utilized a spatial–temporal overview and analysis of potential interactions between meteorological factors and found that the number of reported cases in Shaanbei upland plateau decreased gradually over the study period, while the number of reported cases in the Guanzhong basin increased rapidly. In addition, we observed that the peak of incidence and lag period of meteorological factors was 1 month earlier in Shaanbei upland plateau than Guanzhong basin. Therefore, we assumed that the Guanzhong basin has become a new endemic region of brucellosis, and imported cases have led to the occurrence of hysteresis.

Figure 4 Observed and predicted numbers of human brucellosis cases (A), and the scatterplot (B) in Shaanbei upland plateau, 2005–2018.

Figure 5 Observed and predicted numbers of human brucellosis cases (A), and the scatterplot (B) in Guanzhong basin, 2005–2018.

Figure 6 Relationship between number of human brucellosis cases and temperature (A, B), relative humidity (C, D), rainfall (E, F), sunshine duration (G, H), and evaporation (I, J) in different regions.

Blue lines represent data for Shaanbei upland plateau, and red lines represent data for Guanzhong basin.

In addition to the geographic difference, we observed seasonal fluctuation in the human brucellosis in the study (Fig. 2 and Fig. S2). Consistent with the previous finding that meteorological factors play a crucial role in the seasonal fluctuation and transmission of human brucellosis between reservoir and susceptible populations (Li et al., 2013; Zhao et al., 2019; Cao et al., 2020), our study found that air temperature, sunshine duration, relative humidity, rainfall, and evaporation were associated with the seasonal fluctuation and geographic variation in the incidence of human brucellosis. The metrological factors may influence the dynamics of reservoirs and viral transmission within a susceptible population; other factors such as types of land use, vegetation, and patterns of agricultural production are also possible contributors accounting for the observed association (Fig. S3).

Short sunshine duration and low relative humidity may have contributed to the human brucellosis incidence through a few pathways. First, appropriate sunshine duration and relative humidity have significant effects on estrus of reservoirs, which is critical to the transmission among susceptible herbs of brucellosis (Li et al., 2013; Cao et al., 2020). Second, most animals infected with brucellosis do not show obvious signs, and transmission to the susceptible population occurs mainly through abortion and secretion infection. With the arrival of the production season, a large number of Brucella are excreted into the environment through the abortive secretion of reservoirs. Meanwhile, a large number of susceptible lambs are bred and a dry environment suitable for transmission of Brucella is established, which may lead to higher risks of exposing to susceptible livestock and humans (Yang et al., 2018). In addition, low rainfall and suitable evaporation have shown to exert negative impacts on vegetation growth that serves as food for livestock (Cotterill et al., 2018). Therefore, our findings may suggest that vegetation plays an important role in the transmission and variation of geographical distribution of human brucellosis.

We hypothesized that several factors contributed to the observed trends in human brucellosis cases. Shaanbei upland plateau has been a traditional endemic area for human brucellosis and has experienced several outbreaks. Thus, public and governmental sectors may have strengthened awareness of prevention and have adopted effective and targeted preventive measures. For example, they may have promoted prevention and control of brucellosis in the public and adopted strengthened quarantine practices in endemic areas. As a result, all these measures may have led to the blockage of the transmission routes (Li et al., 2017), the reduction in the incidence, and the reduction of the impact of indirect factors such as meteorological factors. However, the Guanzhong basin is a new endemic area for human brucellosis, and thus, local sectors may have lacked experience and awareness in the prevention and control of the epidemic. In addition, transportation and farming are active in this area, which may have influence on the seasonal fluctuation. In the winter, the livestock and production activities closely related to infected livestock such as cows and sheep are less active. In comparison, in the warm spring and summer seasons, the activities related to farming and production (e.g., lamb delivery, lamb breeding, handling aborted placenta, shearing wool, and processing and trading meat products) that are closely related to infected reservoirs may increase the exposure frequency of the local population to the virus existing in infected reservoirs and contaminated products. Furthermore, high air temperature has been shown to have a positive effect on the persistence and transmission of Brucella (Lee et al., 2013), which is likely to infect susceptible herbs and cause epidemics when protection and awareness are insufficient.

Some limitations of the current study merit consideration. Although the data quality from the National Notifiable Disease Surveillance System is expected to be highly credible, underreporting of cases due to mild or unnoticeable clinical symptoms may still exist. In addition, the forecast model constructed in the current study was based on meteorological factors only and used to predict short-term incidence. Third, we did not have data on other factors that may influence the transmission of human brucellosis such as dietary habits, tourism, and traveling (Zhu et al., 2017); thus, residual confounding factors may exist that could affect the interpretation of the results. Finally, it is necessary to confirm part areas of Guanzhong basin whether has become endemic region of brucellosis through testing of susceptible animals in key areas.

Conclusions

The current study implies that meteorological factors may make important contributions to the transmission and variation in geographical distribution of human brucellosis by affecting the external environment, susceptible livestock and/or human populations, activity of reservoirs or the susceptible population, and vegetation. The established GAMM forecast model is shown to be accurate and applicable for predicting the seasonal fluctuation of human brucellosis. Further studies are warranted to validate our findings and develop reliable and cost-effective prevention strategies to combat human brucellosis.

Supplemental Information

Figure S1 Study site of Shaanxi Province

The map was created by Zurong Yang in ArcGIS 10.1 Software, ESRI Inc., Redlands, CA, USA, (https://www.arcgis.com/index.html).

Click here for additional data file.

Figure S2A The time series distribution of monthly meteorological factors and human brucellosis in Shaanbei upland plateau

Click here for additional data file.

Figure S2B The time series distribution of monthly meteorological factors and human brucellosis in Guanzhong basin

Click here for additional data file.

Figure S3 Cross correlation coefficients between climatic factors and the number of human brucellosis in Shaanbei upland plateau (A) and Guanzhong basin (B), 2005–2018

Cases: monthly number of human brucellosis; Temp: monthly mean temperature; Rain: monthly mean rainfall; R.H: monthly mean relative humidity; S.D: monthly cumulative sunshine duration; Evp: monthly cumulative evaporation; L.C: lag of monthly number of human brucellosis; L.T: lag of monthly mean temperature; L.R: lag of monthly mean rainfall; L.R.H: lag of monthly mean relative humidity; L.S.D: lag of monthly cumulative sunshine duration; L.E: lag of monthly cumulative evaporation.

Click here for additional data file.

Figure S4 Interactions of two meteorological factors and human brucellosis

Click here for additional data file.

Data S1 Raw data of human brucellosis and meteorological factors in Shaanxi Province, northwestern China

Click here for additional data file.

We are grateful to every medical staff struggling on the brucellosis prevention and control, especially those who have contributed to the reporting of human brucellosis in Shaanxi Province.

Additional Information and Declarations

Competing Interests

Author Contributions

Data Availability

The authors declare there are no competing interests.

Zurong Yang and Miaomiao Pang performed the experiments, prepared figures and/or tables, authored or reviewed drafts of the paper, and approved the final draft.

Qingyang Zhou and Shuxuan Song performed the experiments, authored or reviewed drafts of the paper, and approved the final draft.

Weifeng Liang, Junjiang Chen and Tianci Guo analyzed the data, authored or reviewed drafts of the paper, and approved the final draft.

Zhongjun Shao conceived and designed the experiments, authored or reviewed drafts of the paper, and approved the final draft.

Kun Liu conceived and designed the experiments, prepared figures and/or tables, authored or reviewed drafts of the paper, and approved the final draft.

The following information was supplied regarding data availability:

Raw data are available in the Supplemental Files.

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
