# Peer review of "Spatiotemporal expansion of human brucellosis in Shaanxi Province, Northwestern China and model for risk prediction"

_PeerJ, doi:10.7717/peerj.10113_

## Round 0.1 · original submission · Minor Revisions

The reviewers and I agree that your manuscript is well-written and requires only minor revisions to be acceptable for publication. Please see the detailed reviewer comments for these changes. Perhaps the most significant revision is to remove Table 4, which has only a single row, and report those results directly in the text.

·

Basic reporting

No comment

Experimental design

No comment

Validity of the findings

This is an interesting and important paper. In terms of content it is well written and makes an important contribution to the body of work on brucellosis and public health.

Additional comments

The manuscript is well written. Some grammar, however needs to be checked: For example:
Line 26, grammar – with more than 90% of these cases were reported in the northern – “were” needs to be removed
Line 52 – please remove “allergic” as this brings a different connotation
Line 54 and 55 – what is common about clinical presentation of brucellosis is the undulant fever and not necessarily asymptomatic as the author puts it. Please revise this
Line 62 – please replace animal husbandry to livestock keeping.
Line 63 -64 – there is problem with grammar, “including many countries in Latin America, Middle East, and South and Central Asia, which occurred more than 64 500,000 cases every year”. You may consider breaking this from line 62 at livestock keeping.
Line 65 – “with more than 90% of these cases were reported in northern” – remove “were”
Line 66-67 – the author mentions the: apparent geographic expansion” but does not follow up to say a bit more about what this is. So the statement is just hanging

·

Basic reporting

The authors conducted a retrospective epidemiological study in Shaanxi Province from 2005 to 2018 to explore the association between meteorological factors and transmission of human brucellosis according to differences in geographical distribution and seasonal fluctuation in northwestern China. In China, the affected area of brucellosis did expand from northern pastureland provinces to the adjacent grassland and agricultural areas, then to southern coastal and southwestern areas. The findings may be used to predict the seasonal fluctuations of human brucellosis and to develop target prevention and control strategy in the study area.

Experimental design

Line 116, in the part of 'Data collection', given that there are several temperature indicators provided by the Chinese Bureau of Meteorology (http://data.cma.cn/), the authors should indicate the exact temperature index involved, e.g., average air temperature or maximum air temperature.

Validity of the findings

Only one row of data cannot form Table 4, thus there is no need to indicate the GAMM model result in the format of table, text description is enough.

Additional comments

The following typos should be corrected.
1) Line 171, 'Zizhou country' should be 'Zizhou county' ;
2) Lines 187, 244 and the legend of Table 3, 'lag times' should be 'lag time';
3) The article number for reference [2] should be added.
4) Title page, the authors should check the second affiliation of First author, now it is indicated as 'Centre for Disease Prevent and Control ……',is here 'prevent' should be 'prevention'?

---

## Round 0.2 · accepted · Accept

Congratulations on a very nice manuscript, and thank you for your patience with the review process. I look forward to seeing this work in print!

·

Basic reporting

All of my concerns have been addressed properly in the revised manuscript.

Experimental design

no comment

Validity of the findings

no comment

Additional comments

According to my knowledge of PeerJ, the references should be arranged in alphabetical order of the first author of each article.